# How to evaluate a multi-country implementation-focused network: Reflections from the Quality of Care Network (QCN) evaluation

Gloria Seruwagi[1,2]*, Mike English[3], Nehla Djellouli[4], Yusra Shawar[5], Kasonde Mwaba[4], Abdul Kuddus[6], Agnes Kyamulabi[2], Kohenour Akter[5], Catherine Nakidde[1], Hilda Namakula[1], Mary Kinney[7], Tim Colbourn[4], QCN Evaluation Group[¶]

1 Department of Health Policy Planning and Management, School of Public Health, Makerere University, Kampala, Uganda, 2 Department of Social Work and Social Administration, School of Social Sciences, Makerere University, Kampala, Uganda, 3 Centre for Tropical Medicine and Global Health, University of Oxford, Oxford, United Kingdom, 4 Institute for Global Health, University College London, London, United Kingdom, 5 International Health, John Hopkins University, Baltimore, Maryland, United States of America, 6 Perinatal Care Project, Diabetic Association of Bangladesh, Dhaka, Bangladesh, 7 School of Public Health, University of the Western Cape, Cape Town, South Africa

¶ Membership of the QCN Evaluation Group is listed in the Acknowledgments.
* gloria.seruwagi@mak.ac.ug

**Data Availability Statement:** All relevant and available data is included in this paper.

## Abstract

Learning about how to evaluate implementation-focused networks is important as they become more commonly used. This research evaluated the emergence, legitimacy and effectiveness of a multi-country Quality of Care Network (QCN) aiming to improve maternal, newborn and child health (MNCH) outcomes. We examined the QCN global level, national and local level interfaces in four case study countries. This paper presents the evaluation team's reflections on this 3.5 year multi-country, multi-disciplinary project. Specifically, we examine our approach, methodological innovations, lessons learned and recommendations for conducting similar research. We used a reflective methodological approach to draw lessons on our practice while evaluating the QCN. A 'reflections' tool was developed to guide the process, which happened within a period of 2–4 weeks across the different countries. All country research teams held focused 'reflection' meetings to discuss questions in the tool before sharing responses with this paper's lead author. Similarly, the different lead authors of all eight QCN papers convened their writing teams to reflect on the process and share key highlights. These data were thematically analysed and are presented across key themes around the implementation experience including what went well, facilitators and critical methodological adaptations, what can be done better and recommendations for undertaking similar work. Success drivers included the team's global nature, spread across seven countries with members affiliated to nine institutions. It was multi-level in expertise and seniority and highly multidisciplinary including experts in medicine, policy and health systems, implementation research, behavioural sciences and MNCH. Country Advisory Boards provided technical oversight and support. Despite complexities, the team effectively implemented the

**Funding:** This work was funded by the Medical Research Council (MRC) Health Systems Research Initiative 5th call via grant MR/S013466/1 to TC at UCL Institute for Global Health, United Kingdom; YS and JS at Johns Hopkins University, United States of America; KA and AK at Diabetic Association of Bangladesh Perinatal Care Project, Bangladesh; CM at Parent and Child Health Initiative, Malawi; GS at Makerere University School of Public Health, Uganda; and ME at University of Oxford, United Kingdom; and by the Bill & Melinda Gates Foundation via grant INV-007644 to TM at LSHTM, United Kingdom. The funders had no role in study design, data collection and analysis, decision to publish, or preparation of the manuscript.

**Competing interests:** The authors have declared that no competing interests exist.

QCN evaluation. Strong leadership, partnership, communication and coordination were key; as were balancing standardization with in-country adaptation, co-production, flattening hierarchies among study team members and the iterative nature of data collection. Methodological adaptations included leveraging technology which became essential during COVID-19, clear division of roles and responsibilities, and embedding capacity building as both an evaluation process and outcome, and optimizing technology use for team cohesion and quality outputs.

## Introduction

In 2017, WHO and global partners launched 'The Network for Improving Quality of Care for Maternal, Newborn and Child Health' (referred to as the Quality Care Network, QCN) seeking to reduce in-facility maternal, neonatal and stillbirth case fatality rates by 50% within 5 years initially in nine pathfinder countries selected because they had already bought into the quality of care (QoC) agenda and had robust systems in place to support this buy-in: Bangladesh, Cote d'Ivoire, Ethiopia, Ghana, India, Malawi, Nigeria, Tanzania and Uganda [1–3]. Their aim was to encourage cross-country learning on how to implement quality improvement [2]. This would promote partner coordination while emphasising country ownership and leadership, and shared learning.

Though there is an emerging body of work on networks as a potential change strategy this is mostly from high-income countries [4–6] or, on networks focused on drawing attention to global health issues in LMICs [7]. Research on whether and how purposefully created implementation-focused networks such as QCN might leverage global, national and local change is however, sparse [8]. We sought to fill this gap in evidence to inform planning and delivery of future multi-country implementation-focused networks, as well as understand this network and how it could be improved so as to better achieve its goals. We conducted case study work in four out of the nine pathfinder countries (Bangladesh, Ethiopia, Malawi and Uganda). The situation in each of these case study countries is different with respect to political engagement, and on-going and planned activities related to Maternal, Newborn and Child Health (MNCH) that could be leveraged or that present barriers to the successful emergence, legitimacy and effectiveness of QCN in the country.

### An overview of the collection's papers on network emergence, legitimacy and effectivness

This paper is part of a collection of nine papers from the QCN Evaluation project. We asked three broad questions concerning the emergence, legitimacy and effectiveness of the network (Box 1) and these are tackled directly in turn in the other eight papers in our QCN Evaluation collection outlined in S1 Text, which we recommend is read alongside this manuscript. Our collection of 9 papers are the main results from our QCN Evaluation project evaluating the WHO-coordinated Network for Improving Quality of Care for Maternal, Newborn and Child Health.

### QCN arrangements

The QCN was officially launched in 2017 in Lilongwe, Malawi. The QCN's nine pathfinder countries shared key features which include a demonstrated commitment to Quality of Care

Box 1: The QCN evaluation

We examined how QCN is constructed, its operations and their effects. Despite the scale and ambition of QCN, the investments it involved and possibility it could influence the way multilaterals and bilaterals operate in the future, no external evaluation had been commissioned. It therefore needed to be studied. We retrospectively (2016–2018) and prospectively (2018–2022) evaluated what aspects of the QCN work best and how it was influencing global, national, and local levels by tackling three research questions at different levels.

Global level: What attributes of this multi-country network and its operational strategy and performance affect the engagement of QCN actors at global and national levels and their adoption of a shared agenda and goals to improve maternal and newborn health services?

National level: What shapes the relationship between country teams and the global QCN leadership and how does this influence ownership of the policy and management work that is required to set national aims and improve services, and which characteristics of the health system context appear to influence this?

Local level: What specific form does national QCN activity take and how does this influence which specific interventions are delivered, which of these are felt to be successful by local actors and which lead to measurable changes in processes and outcomes?

(QoC) supported by varying existing resources, systems and structures which support providing effective, safe, people-centred care that is timely, equitable, integrated and efficient. It was expected that pathfinder countries would be willing to transparently share data within the network, have a desire to learn and develop, and that international actors and countries would join together to learn from one another, rather than acting in competition. A critical component of this is honesty about gaps in knowledge and areas for improvement.

In terms of composition and roles of the network, the WHO is the global convener, coordinator, and secretariat for the QCN and its learning activities. In addition, one of the WHO's main normative roles is to create technical guidance and standards on QoC and provide support to countries to adapt those standards. The WHO partners with other global actors who in turn implement and fund activities supporting QCN at national and other levels. At the national level for QCN countries, the Ministry of Health (MoH) is the convener with WHO providing technical support. Within QCN countries MoHs and WHO country offices are supported by a plethora of development, implementing and funding partners involved in MNCH including other United Nations, bilateral and philanthropically funded agencies. These partners in turn support implementation of QCN activities at the local level (sub-national and facility).

The work addressing our research questions (Box 1) and reports of how the QCN is organised at all levels from global to facility, country-specific contextual information, detailed methodology and the results of this evaluation can be found in the other papers and supplementary files that are part of this collection (S1 Text, S2 Text, S3 Text). In this paper we investigate, report on, and reflect on our research process, and learning gained from its conduct.

## Methods

### Ethics

Ethical approval was obtained from the Research Ethics Committee at University College London (3433/003) and London School of Hygiene and Tropical Medicine (17541) and the institutional review boards in Bangladesh (BADAS-ERC/EC/19/00274), Ethiopia (EPHI-IRB-240-2020), Malawi (NHSRC Ref,19/03/2264) and Uganda (MAKSPH-HDREC, Ref: 869).

### Methods for investigating, reporting on, and reflecting on our research process

This paper summarises the evaluation team's reflections on this 3.5 year multi-country, multi-disciplinary project. We aimed to examine our approach, consider if we had contributed any methodological innovations, and ask ourselves what lessons we had learned. Our purpose was to produce recommendations for conducting similar research in the future. Methodologically we therefore present an insiders' view of our work. Although planned from early on in the process of the QCN evaluation, this approach is the team's retrospective effort at making sense of how the evaluation unfolded. To inform this process, and in the spirit of co-production, we asked all QCN evaluation team members to reflect on the work they had undertaken and lessons learnt. Additionally, country teams jointly reflected and shared with this paper's first author their thoughts in response to the primary questions, which were shared in form of a focus group discussion (FGD) guide jointly designed by the evaluation team. The FGD guide was administered electronically with country and paper-writing team members by this paper's lead author. Questions asked included 'what worked well?', 'what facilitated the implementation experience in their country?', 'what methodological innovations or adaptations did the team come up with and why?', 'what can be done better?', and 'what recommendations would you give others undertaking similar work?'. Lead authors of all the other QCN-focused papers in this collection also discussed these same questions with their co-authors and shared lessons learnt from developing topic-specific content. Inevitably, there were overlaps in membership of these different groups; evaluation team members, country teams and paper specific research teams. This overlap was valuable in many ways as it partly served to triangulate some key findings. Responses from country and paper-writing teams were analysed thematically across the key themes deduced from the data which were implementation barriers and facilitators, methodological innovations and recommendations for future evaluations. As findings were being refined we then explored published literature on research conduct and learning to contextualise our results.

We first report on the composition, characteristics and organization of our team, and the conduct, phasing, communication and co-ordination of our work. We then present the key themes emerging from our investigation of our research process, including opportunites, methodological adaptations, and researcher experiences and reflections, by country (Table 1).

## Results

### Composition, characteristics and organisation of the evaluation team

First, our team was global—spread out in the four different case study countries. The PI was based in London (University College London–Institute of Global Health / UCL-IGH), along with other key team members. Each of the four evaluation settings had in-country team members affiliated to different institutions, and other members were based outside London and the four cases study countries (Fig 1). The team was highly multidisciplinary and multi-level, with different members having different levels of expertise and seniority. With this complex team

**Table 1. Highlights on key themes from lessons learned.**

| Opportunities | • Rich multidisciplinary theorisation on network complexity<br>• In-team capacity building and support<br>• In-country teams strengthened research credibility and progress |
|---|---|
| **Methodological Adaptations** | • Leveraging technology and virtual working approaches (in Covid-19 context)<br>• Capacity support and division of roles |
| **Researcher Experiences** | • MOH collaboration and possibilities for uptake<br>• Strong support from evaluation team leadership was a major facilitator<br>• Limited [QCN] Network awareness beyond central/national level was a key barrier for data collection at sub-national and facility level |
| | **In- country: Bangladesh**<br>• Previous collaboration with MOH was revitalized, leading to strong support for the evaluation |
| | **In- country: Ethiopia**<br>• An ongoing project (IDEAS) was used to mobilize additional resources for undertaking QCN evaluation work |
| | **In- country: Uganda**<br>• Although costly, a blended approach to data collection addressed low response rate, and stronger networks were established with MOH's new responsible unit |

compositon we were intentional about practically applying key principles of co-production including meaningful collaboration, respect for diversity, empowerment, peer learning; anchored on key values such as inclusivity and transparency–including about how challenging such experiences can be. In each country, the study had an Advisory Board to provide technical oversight and support. Membership of these advisory groups was based mostly on expertise and interest in MNCH, health systems, policy and quality improvement. Membership also represented critical stakeholders in the different countries. For example, in Ethiopia the Senior Advisory Group (SAG) was chaired by the Ministry of Health (MoH), while all in-country QCN stakeholders were members. This is similar to arrangements in the other three countries, and, in addition, there was an international advisory board to support overall implementation of the evaluation.

## Conduct, phasing, communication and co-ordination

The conduct of the evaluation was phased. It began in Malawi and Bangladesh (2019), then Ethiopia (2020) and finally in Uganda (late 2020). Each of these countries conducted repeated phases of data collection, accounting for contextual specificity, but guided by the overarching global QCN framework. Despite implementation complexity, and research conducted as the global COVID-19 pandemic unfolded, the team was able to effectively implement the evaluation. This included considerable cross-learning between countries over time facilitated by many virtual meetings and secure online document sharing. Communication and coordination, supported by specific personnel in the central and national research teams, were key to this evaluation, as were other factors highlighted below.

## Opportunities

Working as a multi-country, multi-disciplinary team presented several opportunities to members which include but are not limited to the following:

1. The ability to theorise from multiple disciplinary angles and then harmonise this theorisation allowed for actionable and effective strategies of implementing the evaluation. The team comprised of experts from different fields including medicine and other clinical disciplines, health systems, implementation research, policy, behavioural sciences, social sciences and others working in the area of maternal, newborn and child health (MNCH).

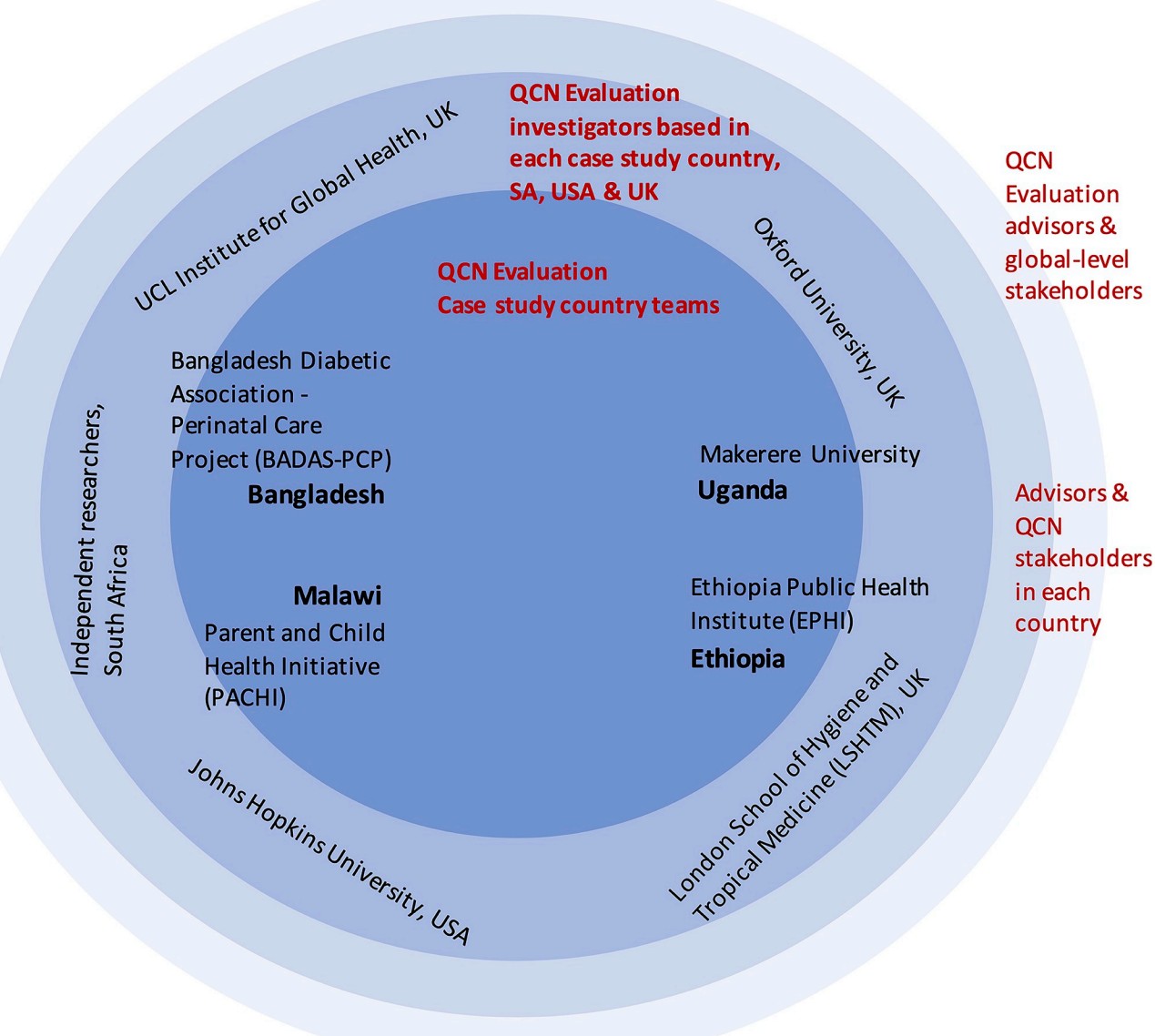

**Fig 1. QCN Evaluation team structure.**

Some were pure academics while others had a blend of research, policy and practice experience. This formed a rich resource pool from which to conceptualise and effectively implement this evaluation. Some of the evaluation team members are renowned global experts in the field of MNCH, health systems, network emergence and effectiveness and conducting global evaluations. Drawing from their expertise and that of other members, this evaluation's rigour was strengthened at all phases right from conceptualisation to completion. This expertise was harnessed through ongoing mentorship and communication, peer reviews, provision of and signposting to relevant resources. Senior members of the team supported junior and mid-level members which positively impacted evaluation quality, speed and team cohesion. The evaluation used a variety of methods and on occasions

specifically mixed methods approaches; it was therefore helpful to have a multi-disciplinary team able to support both qualitative and quantitative methods, including a series of iterative rounds of mixed methods data collection.

2. The approach of collecting data in a series of iterative rounds enabled teams to build rapport with key stakeholders and identify key systems as well as processes. Through multiple rounds of data collection the identified key stakeholders were engaged as direct study participants, gatekeepers or resource persons with links to the additional study participants. The iterative nature of the evaluation also enabled teams to assess progress and refine tools, particularly the qualitative tools, for the next round of data collection. Beyond developing more robust tools, which continued to evolve iteratively, key lessons were also learnt on the actual process of data collection and analysis–including which frameworks and theories would be more valuable in making sense of the data.

3. Capacity building and support for younger researchers was another added advantage. Embedded within this study was capacity building and strengthening as one of its key processes and outcomes. Fidelity to this cause was ensured through selecting a blend of junior, early-career and senior researchers. Intentional mentorship and training sessions were conducted for the teams, such as in NVivo coding and Stakeholder Network Analysis. We also shared, reviewed and made suggestions for improvement on each other's work (for example on papers and country reports). Standard operating protocols, manuals and other necessary tools were shared across in-country teams including practical resources such as timely renewal of licences for data analysis software. Prior working relationships of the in-country teams and some external members were helpful to build on these collaborations. The work further strengthened not only capacity at the individual level, but also consolidated partnerships at the institutional and global level.

4. Having in-country teams strengthened research progress and credibility, mostly because they also knew the network of actors and context better than external researchers, and was critical to the successful conduct of this research during the COVID-19 pandemic. In-country evaluation teams were led by local institutions and researchers, which made it easier for obtaining administrative and ethical clearances. All teams had some sort of relationship (some had written Memorandum of Understanding (MOUs) with the key institutions such as the MoH as well as QCN partners, including implementing partners and funders. These would later seamlessly support the process of data collection within their own institutions and those of their partners, including at the frontline (e.g. health facilities or sub-national authorities). This included: conducting observations in health facilities or meetings within MoH and other related QCN events such as training or supportive supervision; having access to key documentation; and jointly determining which in-country study sites fit the criteria for best and least performing health facilities. It was also easier to harmonise the evaluation schedule with critical in-country processes and milestones. The better understanding of and adaptation to local geopolitical and sociocultural dynamics in the different countries was of particular value. These in-country processes were supported overall by the global team led by the PI, and–as mentioned earlier–communication and coordination were pivotal to the success of the evaluation process.

## Methodological adaptations and ways of working

In order to realise the benefits or advantages of this multi-disciplinary and multi-country team; while being cognizant of the complexity surrounding this evaluation, the team devised several mechanisms to assure success. These included:

1. Virtual working methods: The QCN evaluation team leveraged technology to communicate, support one another, collect, analyse and write up study findings. There was a pre-planned schedule for periodic team calls for all members. These were frequent and done weekly, bi-weekly or monthly depending on the evaluation phase and need. The purpose of these calls was to provide support for all teams, take stock of progress and identify gaps or in-country team needs that needed to be addressed e.g. skills gaps for field or analysis teams. Follow up and action points would be derived from these calls; for example, providing additional training, logistical or technical support. With different members spread out across the world, the Advisory Board meetings were also conducted virtually. These meetings served as accountability and support systems for the evaluation team.

   - The availability of a virtual option prepared the team on how to circumvent the restrictions posed by the COVID-19 pandemic, most of which meant that field teams would not travel for data collection. A lot of data in the final round was collected online (including interviews); but even without COVID-related restrictions, some of the key informants who needed to be interviewed were extremely busy, or were mobile officials, and preferred the virtual option. Moreover, the evaluation's survey arm was also designed to be completed online. The team's early adoption of virtual options via Zoom meetings from the first round of data collection enabled them to ensure continuity, interface with critical stakeholders, experience relatively less disruption and meet data collection targets.

   - Most of the meetings, and some interviews, were held via Zoom. For interviews conducted via Zoom, the team was able to obtain quick and free transcription services as Zoom software provides transcription–though this support was only available for interviews done in English language. Related to this, NVivo shared projects, and use of shared folders for online working across seven countries, encouraged real time and ongoing valuable contributions from the different partner organisations.

   - Linked to this, there was a social norm shift with COVID whereby remote interviews previously were not as common but due to COVID restrictions, they became more acceptable even after restrictions were lifted because people saw cost-saving on their time and access.

2. Capacity building: As earlier highlighted, capacity building was both a process and outcome of this evaluation. It was also a deliberate strategy used to effectively implement this evaluation. Multi-country trainings were conducted online. Training topics included qualitative methods, including analysis, for field teams and any other member who needed it. This later helped in task sharing especially at the analysis stage, when a lot of data was collected and it needed coding and analysis. Many team members/in-country researchers were able to simultaneously support this process and also provide a less biased view about the data assigned to but not collected by them. This was also a professional bonding and cohesive process which made the team stronger and added rigour to the evaluation process.

3. Division of roles: The division of roles and responsibilities did not stop at data analysis; it extended to the latter stages of evaluation and in particular the writing of manuscripts. Training, mentoring and continuously supporting the junior and early-career team members was emphasized. Early-career team members were encouraged to lead and support planned publications with the support of senior members of the team. As testament to this, we have a different first author for each of the nine papers, spread around the different evaluation institutions and countries—several of whom are writing for the first time as first author of a research article. This is another deeply beneficial outcome of mentorship and

support as different members and teams voluntarily took up leadership and support roles in the manuscripts emerging out of this work so far.

## Researcher experiences and reflections: Examples from each country

The evaluation case studies were in Bangladesh, Ethiopia, Malawi and Uganda. Despite their differences, country teams reported some similar experiences in conducting this evaluation. These included how previous collaboration with Ministry of Health enormously supported seamless implementation of the evaluation. The excerpts below show that ongoing relationships with respective health ministries were instrumental in obtaining not only study clearance but also interest, buy-in and active support for study implementation:

*The Diabetic Association of Bangladesh (BADAS) has been working for many years in collaboration with different departments and ministries of Government of Bangladesh including the Ministry of Health and Family Welfare. Through this long-term collaboration, BADAS has gained a firm foothold and could convince the government that to evaluate the Network would be beneficial* (BADAS, Bangladesh Partner)

*Strong collaboration with government (MoH) enabled smooth operation of the research project; starting from the very beginning the MoH offered support letters for the project . . . and the partnership with the government was very helpful in each stage of project implementation* (EPHI/LSTHM, Ethiopia partner)

*Once the MoH contact introduced us to others it was very easy to get access to key teams, partners, data and Network-related events. This was partly because the in-country institution Makerere University School of Public Health (MakSPH) already is a longstanding MOH partner. . . and MakSPH was leading the Learning component of the Network in Uganda* (MakSPH, Uganda partner)

Lack of awareness by frontline workers about QCN initiatives was a key theme in the data; and healthworkers at the local/facility level did not seem to know much about QCN, despite implementing its activities. Related to this was an apparent disconnect between QCN at the central hub (MoH and national level partners) where a lot of planning and activities were routinely happening; and the lower levels which seemed relatively naïve, implementing whatever guidance was passed on to them as either Quality Improvement (QI) or MNCH initiatives:

*The lack of knowledgeable staff about the QCN initiative was very challenging during the data collection, especially at the sub-national and facility level context* (EPHI, Ethiopia)

As shown in the excerpt above, this lack of awareness hampered evaluation efforts in a sense that would-be participants were unable to answer critical questions. It was also difficult, in some countries, to quickly identify participating QCN facilities at the initial stage of data collection. While this lack of awareness had significant methodological implications including participant recruitment and sample size, this challenge enabled the evaluation team to reflect, learn and implement necessary modifications to their evaluation approach, including new participant selection in consultation with key local stakeholders such as the Ministry of Health.

All country research teams were grateful for the support provided by study leadership at UCL-IGH and credited it to enabling them to successfully implement the evaluation:

*We were lucky enough to get all sorts of guidance and support from the UCL team. This includes guidance on topics and tools like survey questionnaire and analysis* (BADAS, Bangladesh)

COVID-19 disrupted study timelines and activities in all case study countries although the teams at global and country level jointly worked quickly to mitigate this potentially negative impact as illustrated in the quote below:

*Due to COVID-19 prevention and control measures, it was difficult to find some partner organisations in office . . . the timing of implementation varied across different implementers and some of the partners withdrew due to different reasons . . . some interviews were conducted virtually and not face-to-face* (EPHI, Ethiopia)

In addition to similarities across the researcher experiences, there were some unique researcher experiences, according to country.

In Bangladesh, the evaluation processes built new partnerships while also consolidating existing collaborations and strengthening accountability between researchers and key stakeholders. Some of these stakeholders were direct participants with whom the team built rapport which they would rely on to collect data and engage at the different data collection rounds. Other stakeholders were institutions and offices with supervisory and oversight support functions who initially only needed the team to demonstrate compliance but would later support the entire process as formidable allies. For example, BADAS in Bangladesh signed an MOU with the Health Economics Unit (HEU) of the Ministry of Health and Family Welfare. This MOU would later be a point of reference for MoH's support which enabled BADAS to seamlessly obtain enhanced access to key meetings, partners, files or documents and MNCH related activities.

In Ethiopia, the team benefitted from having the direct support and partnership of the London School of Hygiene and Tropical Medicine (LSHTM). LSHTM was already in partnership and implementing related projects with the Ethiopian Institute of Public Health (EPHI) through the IDEAS project [9]. Leveraging this partnership, they were able to obtain funding for Ethiopia to join the QCN evaluation case study countries which were originally three (Bangladesh, Malawi and Uganda). This provided an additional study country and Ethiopian work added considerable value to the evaluation in terms of health system experiences, achievements and the additional skills offered by their academic partnership with LSHTM.

In Uganda, the response rate to the online survey was extremely low. This was at all levels but especially from the frontline workers at health facilities, mostly because of heavy workloads and technology constraints. The survey would therefore not have achieved the participant numbers needed from the different levels of the in-country QCN stakeholders. The team, with support from the PI and other members, addressed this challenge by adopting a blended approach–including embedding a manual component and hiring additional field team members to administer hard copy questionnaires at health facilities whose data would later be uploaded onto the online platform. Although this method was more costly and time consuming, it increased participation by a significant margin (> 75%). Furthermore, it opened the overall (global) evaluation team to in-country contextual challenges that were also affecting and mirroring the implementation of QCN work.

## Discussion

The overall evaluation aimed to investigate how the QCN network functions–i.e. whether it was 'working'—at global level, at national level, and at local level in selected health facilities in the case study countries. This paper focused on how the team was able to undertake this multi-country, multi-year, multi-disciplinary and phased evaluation of a complex global network for maternal and child health. We draw lessons lessons from the set up, functioning and outcomes

of the evaluation processes–the other eight papers in this collection address evaluation specific questions.

## What went well

Strong study leadership was the most critical success factor for this multi-country, multi-disciplinary evaluation. The team at UCL-IGH devoted considerable resources and effort to bring everyone together at all stages, this was critical for success. Membership selection of the overall team, for countries and individuals, was strategic and supportive. The leadership is credited for enabling an extremely high level of participation and in-country led processes of adapting the evaluation in each case study country. Team member mix of senior, mid- and early-career members from different disciplines was instrumental in achieving the capacity strengthening function; while also readily supporting future work on the project such as analysis and manuscript development. Team members were highly motivated, working hard towards achieving the shared goal while also learning from one another.

These listed enablers for successful study implementation have been applied and cited elsewhere; for example Nuttal et al's [10] experience of carefully selecting and supporting local key contextual leads; undertaking rigorous back translations and using information systems for monitoring; as well as ensuring strong cohesive leadership at the centre as a multi-pronged innovative response to the challenges encountered. Standardising and having clear, robust protocols has also been reported as important to successful implementation of large, multi-country studies [11, 12] and networks [13]. The use of virtual data collection methods using online platforms like Zoom have been shown to put some participants more at ease than in-person interaction [14] although several other challenges have also been noted. These include the fact that these methods are not suitable or appropriate for some research topics or participants; there can be limited opportunity for researchers to provide support or empathy; restrictions on researcher ability to assess the context, non-verbal communication or body language; exclusion of potential participants without access to computers or internet; technical or internet challenges and security [15–17]. Personal and in-country connections and previous collaboration among team members–especially the senior ones–were critical success factors for this evaluation, also noted elsewhere [18]. It is worth highlighting that the lessons above speak to both evaluation teams and network collaborations.

Adopting a co-production approach in the conceptualisation, design and implementation of this evaluation was a strong enabler. Adeptly captured in the Thai concept of the *"triangle that moves the mountain"* whereby researchers, citizens, and policy makers work together to achieve change [19–21]. Co-production has gained traction in the last decade because of its potential to improve the quality and relevance of research and its effect on policy and practice [21–23]. It continues to increasingly inform health decisions [21, 24, 25]. The QCN evaluation drew from the co-production principles of being context-based, pluralistic, goal-oriented and interactive [26]. Using these strategies the study team was able to collect robust data and seamlessly navigate what could otherwise have been relatively closed in-country terrains; it also secured the critical support of key stakeholders like Ministries of Health with what seems a greater appetite for research uptake.

A growing body of evidence continues to highlight the invaluable principles, processes and outcomes or benefits of co-production to include the mobilisation of multiple knowledge sources within a given setting and re-imagining the power dynamics of knowledge production as well as the support derived from long term collaboration and complex trust-based relationships [27–29] with clear linkages on how co-production processes can support improved health system decision making practice [24, 30]. The challenges of co-production have been

widely acknowledged including conflict, resource and time constraints, although these have also been associated with positive outcomes for example by increasing productivity in the longer term [22, 26, 31]. More importantly, co-production is well aligned with the current push for decolonization of global health [32–34] whose prominence has significantly increased in the pandemic era.

## Lessons learnt

The use of multi-method approaches and subsequent triangulation was invaluable in strengthening the rigour and validity of evaluation results. Successful application of the diverse evaluation methods was partly contingent on contextual diversity across countries, the realities of conducting research amidst COVID-19 pandemic restrictions and improvisation of in-country teams among others. Other lessons are shared throughout this paper.

## What can be done better and recommendations for undertaking similar work

Stronger resource mobilisation and alternative funding mechanisms: Teams undertaking such work should be better resourced—specifically in terms of funds, although other resources remain important. The study team had excellent institutional and member partnerships with strong expertise. However, like many other such projects, the evaluation was resource constrained. This became more pronounced because of the complexity entailed in conducting the project: the work was multi-country, multidisciplinary with multi-level membership in terms of seniority. Phased implementation with multiple rounds of data collection was employed that was affected by the COVID-19 pandemic which also had resource implications. Generally, COVID-19 led to the diversion of funds and a number of funding agencies re-allocated and restricted available funding for the Global South (for example the UK Government ODA fundings cuts). What were previously limited resources were therefore even more spread thinly. Additional funding was needed and obtained through IDEAS-LSHTM, but this was earmarked for evaluation only in Ethiopia. Furthermore, as with all grants there were some conditions attached, which also affected implementation to some extent. For example, as lead institution UCL was, as per funding instructions, unable to remit additional resources to different country teams which needed them (such as those saved on travel from pandemic-related restrictions which would have financed in-country activities). Additionally, while a short no-cost extension was received, the funding agency required that all remaining funds (unspent due to delays caused by the pandemic) were spent within a 3 month window period. These resources could not be given to partner institutions in the Global South. All in all, the conduct of this evaluation highlighted the need for ambitious but important projects to have sufficient budgets and budget flexibility to promote success. We note again the substantial funding devoted to QCN activities did not include an external evaluation. This need for alternative funding approaches to support co-production and complex yet effective partnership working arrangements like the QCN Evaluation continues to be highlighted in recent literature [22, 27], some with calls to fund "partnerships rather than projects" [35]. As some agencies start or continue to establish funding opportunities that support long term relationships and co-production we can expect to harness related benefits which ultimately contribute to shared goals such as improving maternal, newborn and child health or stronger health systems [36].

There is enthusiasm about the results of this evaluation from stakeholders, such as WHO and Ministries of Health in all case study countries, who were actively involved in the process. This holds a lot of promise and potential for one aim of the QCN, "learning". Under the leadership of the participating countries' health ministries, QCN aimed to support quality

improvement by pursuing four strategic objectives: leadership, action, learning and account-ability (LALA) [2, 37]. There have been strong calls for "learning health systems" [36, 38, 39], where learning is understood to mean "the ability to gather and use relevant knowledge and intelligence to bring about improvements in performance" [36] which align with the objectives and intended outcomes of this evaluation. Our QCN evaluation was independently funded, adopted an implementation approach informed by co-production and seems to have resulted in key stakeholders like WHO and in-country partners led by MOH being keen to see and hopefully use the results. Ideally, our research uptake strategy would be more ambitious and show a clear impact pathway–beyond disseminating to WHO/MOH and their immediate insider coalition networks including implementing partners. This would require longer-term support enabling evaluation teams to undertake additional knowledge translation and advocacy work that is rarely sufficiently supported by time-limited research funding. Such extended work to bridge the gap between research, policy and practice remains a major challenge for most LMICs; and if the 50% reduction target for MNCH is to be realised, stakeholders in the three worlds will need to work differently. Creating long-term learning partnerships could be one solution.

## Limitations of this paper

The process we describe has limitations. With an insider view, a potentially more critical outsider perspective is lacking and our long-term collaborative work may result in some assumptions or ideas being shared within and across countries and team members. Our results therefore represent a particular interpretation of events developed with the pragmatic intent of informing future similar exercises, with strengths of our reflective methods outlined in earlier sections.

## Conclusion

Our research has generated useful evidence to support future global networks, important theoretical insights into how networks operate and how operations could be improved to directly benefit maternal, newborn and child health. These specific forms of learning are reported in the other eight papers that are part of this collection. This paper reports what we have learned from the experience of conducting a rigorous evaluation employing a multi-country network of researchers conducting a complex, inter-related set of projects. Key elements included the need for balancing standardization with in-country adaptation, strong leadership from the centre, optimizing technology for team cohesion and a focus on expert supported capacity development to deliver quality outputs. We believe the partnership approach taken is a useful model and enables such a range of work to be undertaken as exemplified by this collection's different papers while having the potential to deliver the learning that will be necessary to help the global community achieve goals of reducing maternal, newborn and stillbirth case fatality by 50%.

## Supporting information

**S1 Checklist.**
(DOCX)

**S1 Text. PLOS global public health QCN evaluation collection 2-page summary.**
(DOCX)

**S2 Text. QCN papers common methods section.**
(DOCX)

**S3 Text. QCN papers common country context.**
(DOCX)

## Acknowledgments

¶ The QCN Evaluation Group is: Nehla Djellouli, Kasonde Mwaba, Callie Daniels-Howell, Tim Colbourn (UCL Institute for Global Health, UK), Kohenour Akter, Fatama Khatun, Mithun Sarker, Abdul Kuddus, Kishwar Azad (BADAS-PCP Bangladesh), Kondwani Mwandira, Albert Dube, Gladson Monjeza, Rachel Magaleta, Zabvuta Moffolo, Charles Makwenda (Parent and Child Health Initiative, Malawi), Mary Kinney, Fidele Mukinda (independent researchers, South Africa), Mike English (Oxford University), Yusra Shawar, Will Payne, Jeremy Shiffman (Johns Hopkins University, USA), Catherine Nakidde, Agnes Kyamulabi, Hilda Namakula, Gloria Seruwagi (Makerere University, Uganda), Anene Tesfa, Asebe Amenu, Theodros Getachew, Geremew Gonfa (Ethiopia Public Health Institute, Ethiopia), Seblewengel Lemma, Tanya Marchant (LSHTM, UK).

## Author Contributions

**Conceptualization:** Gloria Seruwagi, Nehla Djellouli, Tim Colbourn.

**Data curation:** Gloria Seruwagi, Nehla Djellouli, Tim Colbourn.

**Formal analysis:** Gloria Seruwagi, Mike English, Yusra Shawar, Kasonde Mwaba, Agnes Kyamulabi, Kohenour Akter, Catherine Nakidde, Hilda Namakula, Tim Colbourn.

**Funding acquisition:** Gloria Seruwagi, Nehla Djellouli, Abdul Kuddus, Mary Kinney, Tim Colbourn.

**Investigation:** Tim Colbourn.

**Methodology:** Gloria Seruwagi, Nehla Djellouli, Tim Colbourn.

**Project administration:** Gloria Seruwagi, Nehla Djellouli, Abdul Kuddus, Tim Colbourn.

**Resources:** Mike English, Nehla Djellouli, Tim Colbourn.

**Software:** Nehla Djellouli, Tim Colbourn.

**Supervision:** Mike English, Nehla Djellouli, Abdul Kuddus, Tim Colbourn.

**Validation:** Gloria Seruwagi, Agnes Kyamulabi, Catherine Nakidde, Hilda Namakula, Tim Colbourn.

**Visualization:** Gloria Seruwagi, Tim Colbourn.

**Writing – original draft:** Gloria Seruwagi.

**Writing – review & editing:** Gloria Seruwagi, Mike English, Nehla Djellouli, Yusra Shawar, Kasonde Mwaba, Abdul Kuddus, Agnes Kyamulabi, Kohenour Akter, Catherine Nakidde, Hilda Namakula, Mary Kinney, Tim Colbourn.

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
