## [Decision Letter · Decision Letter 0]

11 May 2023

PGPH-D-23-00381

How to evaluate a multi-country implementation-focused network: Lessons from the Quality of Care Network (QCN) evaluation

Dear Dr. Seruwagi,

Thank you for submitting your manuscript to PLOS Global Public Health. After careful consideration, we feel that it has merit but does not fully meet PLOS Global Public Health’s publication criteria as it currently stands. Therefore, we invite you to submit a revised version of the manuscript that addresses the points raised during the review process.

Please note that we have only been able to secure a single reviewer to assess your manuscript. We are issuing a decision on your manuscript at this point to prevent further delays in the evaluation of your manuscript. Please be aware that the editor who handles your revised manuscript might find it necessary to invite additional reviewers to assess this work once the revised manuscript is submitted. However, we will aim to proceed on the basis of this single review if possible.

The reviewer has provided comments requesting clarification of matters described in the introduction, methods and results sections, as well as potential improvement of the discussion section. Please ensure you address each of the reviewer's comments when revising your manuscript; in particular, please note the reviewer's comment regarding the necessity for the methods described in this study not to require access to another publication to understand and ensure your methods section provides sufficient information as required.

We look forward to receiving your revised manuscript.

Kind regards,

Hugh Cowley

Staff Editor

Journal Requirements:

1. Please provide separate figure files in .tif or .eps format only and remove any figures embedded in your manuscript file. Please also ensure all files are under our size limit of 10MB.

2. We have noticed that you have uploaded Supporting Information files, but you have not included a list of legends. Please add a full list of legends for your Supporting Information files after the references list. 

3. In the online submission form, you indicated that "Relevant data has been submitted alongside this manuscript and additional data is available on request". All PLOS journals now require all data underlying the findings described in their manuscript to be freely available to other researchers, either 1. In a public repository, 2. Within the manuscript itself, or 3. Uploaded as supplementary information.

Additional Editor Comments (if provided):

Reviewers' comments:

Reviewer's Responses to Questions

**Comments to the Author**

1. Does this manuscript meet PLOS Global Public Health’s publication criteria? Is the manuscript technically sound, and do the data support the conclusions? The manuscript must describe methodologically and ethically rigorous research with conclusions that are appropriately drawn based on the data presented.

Reviewer #1: Partly

2. Has the statistical analysis been performed appropriately and rigorously?

Reviewer #1: I don't know

3. Have the authors made all data underlying the findings in their manuscript fully available (please refer to the Data Availability Statement at the start of the manuscript PDF file)?

Reviewer #1: Yes

4. Is the manuscript presented in an intelligible fashion and written in standard English?

Reviewer #1: Yes

5. Review Comments to the Author

Reviewer #1: Interesting and important topic. However, in general the paper is difficult to read due to the large amount of abbreviations and overload of information in some paragraphs. The paper needs to be majorly re-structured.

Introduction

Line 71, is not clear what the aim was, too vague wording.

Line 82, references or definition of pathfinder countries is missing

Line 88, this whole paragraph is confusing, what is this paper about? I would shorten this or mention the other papers in a box

Line 133, what do you mean by Quality of Care (QoC) definition?

Parts of what is described in "the QCN Evaluation" (line 151) belong in the method section.

Methods

Are the methods about the content of this paper? Or of the whole project and all the papers?

This has to be made clear. I am also missing some references to different methods used, the reader cannot be required to go to a different paper/supplement to understand the basic methods.

It also seems sometimes results/discussion items are mentioned in the methods, such as line 213 "the team was able to effectively implement the evaluation". And the final part of the methods section, starting at line 234 "The process we describe has limitations"

Results

It is also not clear how the method section leads to the results described. Needs more linkage and the result section needs more context, why are are these results discussed?

Discussion

The discussion should start by going back to original aim of the study. This relates to an earlier comment, it is not clear what this paper is describing. Make sure no "new" results are mentioned in the discussion.

6. PLOS authors have the option to publish the peer review history of their article (what does this mean?). If published, this will include your full peer review and any attached files.

**Do you want your identity to be public for this peer review?** For information about this choice, including consent withdrawal, please see our Privacy Policy.

Reviewer #1: No

---

## [Decision Letter · Decision Letter 1]

2 Nov 2023

PGPH-D-23-00381R1

How to evaluate a multi-country implementation-focused network: Lessons from the Quality of Care Network (QCN) evaluation

Dear Dr. Seruwagi,

Thank you for submitting your manuscript to PLOS Global Public Health. After careful consideration, we feel that it has merit but does not fully meet PLOS Global Public Health’s publication criteria as it currently stands. Therefore, we invite you to submit a revised version of the manuscript that addresses the points raised during the review process.

Your revised manuscript has been evaluated by two reviewers whose comments are available below. Both reviewers express concerns about the lack of clarity around the research question and distinguishing the methods/results/discussion sections more clearly.

Could you please revise the manuscript to carefully address the concerns raised?

We look forward to receiving your revised manuscript.

Kind regards,

Steve Zimmerman, PhD

PLOS Staff Editor

Journal Requirements:

Additional Editor Comments (if provided):

Reviewers' comments:

Reviewer's Responses to Questions

**Comments to the Author**

1. If the authors have adequately addressed your comments raised in a previous round of review and you feel that this manuscript is now acceptable for publication, you may indicate that here to bypass the “Comments to the Author” section, enter your conflict of interest statement in the “Confidential to Editor” section, and submit your "Accept" recommendation.

Reviewer #1: (No Response)

Reviewer #2: (No Response)

2. Does this manuscript meet PLOS Global Public Health’s publication criteria? Is the manuscript technically sound, and do the data support the conclusions? The manuscript must describe methodologically and ethically rigorous research with conclusions that are appropriately drawn based on the data presented.

Reviewer #1: No

Reviewer #2: No

3. Has the statistical analysis been performed appropriately and rigorously?

Reviewer #1: N/A

Reviewer #2: I don't know

4. Have the authors made all data underlying the findings in their manuscript fully available (please refer to the Data Availability Statement at the start of the manuscript PDF file)?

Reviewer #1: Yes

Reviewer #2: Yes

5. Is the manuscript presented in an intelligible fashion and written in standard English?

Reviewer #1: Yes

Reviewer #2: Yes

6. Review Comments to the Author

Reviewer #1: As this is submitted as a Research Article I expect a clear distinction between the method and the result section but also a clear relation between them. It becomes a bit clearer why the results were chosen, but this should have also been mentioned at the beginning of the results section. As the authors indicate themselves the distinction between method and result is not clearly stated as it is more a reflective piece. I do feel that the authors should have another critical look and make sure that the methods only describe methods and no results.

Reviewer #2: This paper is a reflective piece about a 3.5 year multi-country, multi-disciplinary project. This paper is not really about quality of care for maternal health - the paper has almost nothing to say about that - so much as the operational challenges and learnings of organizing a multi-country project. I think there's some worthwhile points in the paper to this effect, but I think it should be presented differently.

As you say in your response to the previous reviewer, this paper doesn't fit well with the standard methods/results/discussion format of scientific reporting. I also agree that it doesn't work well in this format. So why have you chosen to present it this way? I thought the previous reviewer made some good points about the limitations of your manuscript along these lines, but you haven't addressed these points so much as dismissed them as being a consequence of the format you've chosen to adopt.

The methods section is still confusing, as it briefly summarizes the methods for the project as a whole, and then the methods for this paper. The latter isn't very clear. It says the first author asked other team members to reflect on "primary questions". But you don't explain how you conducted this, nor clearly indicate what the primary questions are. By labelling this paper's approach as a method, you invite methodological scrutiny. The limitations to the methods should come later on in a limitations section.

There's also some lack of continuity in this paper. For example, co-production is discussed over two paragraphs in the discussion, but is not in the methods or results sections. The balance between standardization and contextual adaption is mentioned as a key finding in the abstract and conclusion, but this finding isn't sufficiently developed in the results or discussion sections.

Ultimately, I feel that you haven't taken the previous reviewer's comments seriously enough, which is the main reason for my decision as they raised similar concerns to mine. I think a good way forward could potentially be to rewrite this as more of a discussion/commentary piece (i.e. not following the methods/results/discussion format) and reconsider where you submit it.

7. PLOS authors have the option to publish the peer review history of their article (what does this mean?). If published, this will include your full peer review and any attached files.

**Do you want your identity to be public for this peer review?** For information about this choice, including consent withdrawal, please see our Privacy Policy.

Reviewer #1: No

Reviewer #2: No

---

## [Decision Letter · Decision Letter 2]

9 Jan 2024

PGPH-D-23-00381R2

How to evaluate a multi-country implementation-focused network: Lessons from the Quality of Care Network (QCN) evaluation

Dear Dr. Seruwagi,

Thank you for submitting your manuscript to PLOS Global Public Health. After careful consideration, we feel that it has merit but does not fully meet PLOS Global Public Health’s publication criteria as it currently stands. Therefore, we invite you to submit a revised version of the manuscript that addresses the points raised during the review process.

Dear authors, I have chosen major revision as I think that your paper still needs some substantial work. 

First, please consider revising the title of the paper. It suggests that the paper is methodological in nature, advising how to evaluate a network, whereas the content is reflecting on the network. 

Please also revise the abstract. Couch the background section of the abstract into the warrant of the paper - why is this important, in relation to other studies?

Please clarify the methods section again as suggested by reviewers. 

Please add relevant intenational network literautre into the paper, so the reader understands more how this paper furthers the field in understanding such collaborations. 

Kind regards

Salla

Please submit your revised manuscript by . If you will need more time than this to complete your revisions, please reply to this message or contact the journal office at globalpubhealth@plos.org. Please include the following items when submitting your revised manuscript:

We look forward to receiving your revised manuscript.

Kind regards,

Salla Atkins, PhD

Academic Editor

Journal Requirements:

Additional Editor Comments (if provided):

Reviewers' comments:

Reviewer's Responses to Questions

**Comments to the Author**

1. If the authors have adequately addressed your comments raised in a previous round of review and you feel that this manuscript is now acceptable for publication, you may indicate that here to bypass the “Comments to the Author” section, enter your conflict of interest statement in the “Confidential to Editor” section, and submit your "Accept" recommendation.

Reviewer #1: (No Response)

Reviewer #2: All comments have been addressed

2. Does this manuscript meet PLOS Global Public Health’s publication criteria? Is the manuscript technically sound, and do the data support the conclusions? The manuscript must describe methodologically and ethically rigorous research with conclusions that are appropriately drawn based on the data presented.

Reviewer #1: Partly

Reviewer #2: Yes

3. Has the statistical analysis been performed appropriately and rigorously?

Reviewer #1: N/A

Reviewer #2: N/A

4. Have the authors made all data underlying the findings in their manuscript fully available (please refer to the Data Availability Statement at the start of the manuscript PDF file)?

Reviewer #1: No

Reviewer #2: Yes

5. Is the manuscript presented in an intelligible fashion and written in standard English?

Reviewer #1: Yes

Reviewer #2: Yes

6. Review Comments to the Author

Reviewer #1: The method section remains a bit unclear and would benefit from more structure, for example like the descriptions in S2. For example how where the questions asked, by whom and to whom, online, in person, questionnaire, focus group? What analysis did you use? As written now, the method section is not reproducible.

The result section is a lot of text and would benefit from some table and/or figures to clarify which stakeholders where involved and what the process looked like etc. Please check some of the phrases for grammatical mistakes.

Reviewer #2: The paper is a lot more cohesive now and it is easy to see the rationale, method, results, and discussion. You've also addressed the continuity issues I raised.

7. PLOS authors have the option to publish the peer review history of their article (what does this mean?). If published, this will include your full peer review and any attached files.

**Do you want your identity to be public for this peer review?** For information about this choice, including consent withdrawal, please see our Privacy Policy.

Reviewer #1: No

Reviewer #2: No

---

## [Decision Letter · Decision Letter 3]

5 Apr 2024

PGPH-D-23-00381R3

How to evaluate a multi-country implementation-focused network: Lessons from the Quality of Care Network (QCN) evaluation

Dear Dr. Seruwagi,

Thank you for submitting your manuscript to PLOS Global Public Health. After careful consideration, we feel that it has merit but does not fully meet PLOS Global Public Health’s publication criteria as it currently stands. Therefore, we invite you to submit a revised version of the manuscript that addresses the points raised during the review process.

Unfortunately, neither of the two reviewers who had previously reviewed your manuscript were available to review the most recent version. However, we have found an additional external reviewer, and their comments are available below.

The reviewer has raised a number of concerns, including some that were raised by the previous reviewers (e.g., improving the methods section). 

Could you please carefully revise the manuscript to address all comments raised?

We look forward to receiving your revised manuscript.

Kind regards,

Steve Zimmerman, PhD

PLOS Staff Editor

Journal Requirements:

Additional Editor Comments (if provided):

Reviewers' comments:

Reviewer's Responses to Questions

**Comments to the Author**

1. If the authors have adequately addressed your comments raised in a previous round of review and you feel that this manuscript is now acceptable for publication, you may indicate that here to bypass the “Comments to the Author” section, enter your conflict of interest statement in the “Confidential to Editor” section, and submit your "Accept" recommendation.

Reviewer #3: All comments have been addressed

2. Does this manuscript meet PLOS Global Public Health’s publication criteria? Is the manuscript technically sound, and do the data support the conclusions? The manuscript must describe methodologically and ethically rigorous research with conclusions that are appropriately drawn based on the data presented.

Reviewer #3: No

3. Has the statistical analysis been performed appropriately and rigorously?

Reviewer #3: I don't know

4. Have the authors made all data underlying the findings in their manuscript fully available (please refer to the Data Availability Statement at the start of the manuscript PDF file)?

Reviewer #3: No

5. Is the manuscript presented in an intelligible fashion and written in standard English?

Reviewer #3: Yes

6. Review Comments to the Author

Reviewer #3: Thank you for the opportunity to review your work. The manuscript titled "How to evaluate a multi-country implementation-focused network: Lessons from the Quality of Care Network (QCN) evaluation" presents an examination and reflection on the process of evaluating a multi-country Quality of Care Network aimed at improving maternal, newborn, and child health (MNCH) outcomes.

Literature on multi country global networks is generally sparse and therefore important, especially given the continued use of these arrangements. The authors are attempting to tackle a concept that deserves attention.

Please consider the follow reflections for consideration:

1. Clarify the Research Question: The fundamental research question is not clear. On one hand, the manuscript suggests that it is attempting to demonstrate if this evaluation method is better, worse, or equally effective as other evaluation methods. On the other hand, this manuscript is attempting to explain the value of this specific network.

2. Compare this Evaluation to Others: The authors provide detail about what they did, yet there is little comparison of this evaluation method contrasted against others. Incorporating and discussing relevant international network literature could significantly enrich the paper. It would place the QCN evaluation within a broader context, helping to underscore how this paper advances the understanding of evaluating such networks and collaborations.

3. Provide Context about the Network: This manuscript did not provide the reader with any insights to suggest that the network was successful in achieving the goal – lower case fatality rates. The reader should be informed if this manuscript (evaluation of a network) is reviewing a failed or successful network.

4. Strengthen the Methods: Based on the title, this manuscript seeks to provide a “how to” yet the methods section is sparse. In particular, the analytical methods are largely missing in the methods section. The authors explain that in addition to individual country analyses, a series of core questions were asked. The authors explain the team composition and how the team communicated yet it is not clear how the authors understood success or failure of the network.

5. Check Title Clarity: The title could be perceived as broader than the manuscript's content. It may benefit from a revision to directly reflect the paper's essence, which is a reflective analysis of the evaluation process of the QCN.

6. Revisit Innovation Claims: I am skeptical that there was innovation in the ways of working. During the time of this evaluation, the world was undergoing a massive shift towards video/teleconferencing linked to COVID. Against this backdrop, the authors are not clear how this work is innovative.

7. Provide Evidence: The section titled “what went well” includes statements about the network (e.g. co-production) with external citations. However, this evaluation is largely a qualitative analysis, yet the authors do not provide qualitative evidence to back up these statements about what went well.

8. Consider Additional Literature: The authors may strengthen this manuscript by reviewing literature from similar networks that do not seem to be featured here (Joint Learning Network, Salud Mesoamerica network). In addition, there is a robust literature on network theory outside of healthcare which, if leveraged, could ground this analysis more holistically.

7. PLOS authors have the option to publish the peer review history of their article (what does this mean?). If published, this will include your full peer review and any attached files.

**Do you want your identity to be public for this peer review?** For information about this choice, including consent withdrawal, please see our Privacy Policy.

Reviewer #3: No

---

## [Editor Report · Decision Letter 4]

5 Jun 2024

How to evaluate a multi-country implementation-focused network: Reflections from the Quality of Care Network (QCN) evaluation

PGPH-D-23-00381R4

Dear Dr Seruwagi,

We are pleased to inform you that your manuscript 'How to evaluate a multi-country implementation-focused network: Reflections from the Quality of Care Network (QCN) evaluation' has been provisionally accepted for publication in PLOS Global Public Health.

Best regards,

Julia Robinson

Executive Editor